# Describe the morphology and mitochondrial genome of *Mecidea indica* Dallas, 1851 (Hemiptera, Pentatomidae), with its phylogenetic position

Chao Chen[1], Dongmei Bai[1], Zhenhua Zhang[1], Xiaofei Ding[1], Shuzhen Yang[1], Qing Zhao[1]*, Hufang Zhang[2]*

**1** College of Plant Protection, Shanxi Agricultural University, Taigu, Shanxi, China, **2** Department of Biology, Xinzhou Teachers University, Xinzhou, Shanxi, China

\* zh_hufang@sohu.com (HZ); zhaoqing86623@163.com (QZ)

**Data Availability Statement:** All files are available from the NCBI database (accession number: OR654110).

## Abstract

We here describe the external morphology and complete mitochondrial genome characteristics of *Mecidea indica* Dallas, 1851, and clarify the evolutionary rate and divergence time. The *M. indica* mitochondrial genome length is 15,670 bp, and it exhibits a typical high A+T-skew (76.31%). The sequence shows strong synteny with the original gene arrangement of *Drosophila yakuba* Burla, 1954 without rearrangement. The *M. indica* mitochondrial genome characteristics were analyzed, and phylogenetic trees of Pentatomidae were reconstructed using Bayesian methods based on different datasets of the mitochondrial genome datasets. Phylogenetic analysis shows that *M. indica* belongs to Pentaotominae and form a sister-group with *Anaxilaus musgravei* Gross, 1976, and Asopinae is highly supported as monophyletic. Molecular clock analysis estimates a divergence time of Pentatomidae of 122.75 Mya (95% HPD: 98.76–145.43 Mya), within the Mesozoic Cretaceous; the divergence time of *M. indica* and *A. musgravii* was no later than 50.50 Mya (95% HPD: 37.20–64.80 Mya). In addition, the divergence time of Asopinae was 62.32 Mya (95% HPD: 47.08–78.23 Mya), which was in the Paleogene of the Cenozoic era. This study is of great significance for reconstructing the phylogeny of Pentatomidae and providing insights into its evolutionary history.

## Introduction

Pentatomidae is the largest group of species in the superfamily Pentatomoidea and is widely distributed worldwide. Currently, approximately 5000 species and more than 900 genera have been recorded [1, 2]. Most species of Pentatomidae are herbivorous, and many species are considered to be primary crop pests worldwide, causing huge losses every year [2]. Phytophagous species feed on the liquid flowing in the vegetative organs of the host plant through their piercing-sucking mouthparts, causing plants to wither and/or die. They are important agricultural and forestry pests [3]. For example, *Nezara viridula* (Linnaeus, 1758) damages rice;

**Funding:** This research was funded by the National Science Foundation Project of China (No.31872272); the Research Project Supported by Shanxi Scholarship Council of China (Nos. 2020-064), Natural Science Research General Project of Shanxi Province (Nos.202103021224331), Key Forestry Research and Development Plan of Shanxi Province (LYZDYF2023-35). The funders had no role in study design, data collection and analysis, decision to publish, or preparation of the manuscript.

**Competing interests:** The authors have declared that no competing interests exist.

*Halyomorpha halys* (Stål, 1855) damages apples, pears, and other fruit trees; and species of the genus *Eurydema* Laporte, 1833 damages cruciferous vegetables. However, most species of Asopinae (Heteroptera: Pentatomidae) are predatory stink bugs that feed on the larvae of Lepidoptera and Coleoptera and can be used for biological control [4–6].

The genus *Mecidea* (Hemiptera: Pentatomidae) comprises a group of stink bugs that occur in subtropical and adjacent temperate parts of the world. Within these regions, the distribution of the genus appears to coincide closely with that of xerophytic or semi-xerophytic environments [7]. This coincidence was established by Dallas in 1851 for two species, *indica* (Bengal) and *linearis*. *Mecidea indica* is a member of this genus. Sailer reviewed the genus *Mecidea* in 1952, including *M. indica*, and provided species literature, identification keys, descriptions and figures [7]. Hsiao et al. (1977) [8] recorded this species in China, and provided habitus photographs and brief descriptions. Rider and Zheng (2002) [9] updated the distribution of this species in China. Rider (2006) [10] provided the most recent worldwide distribution information on this species. The latest literature on *M. indica* was provided by Fan (2011) [11], who produced a description that lacked genitalia information.

A typical insect mitochondrial genome is a double stranded covalently closed circular DNA molecule, including 37 genes (13 protein coding genes (PCGs), 22 transport RNA genes (tRNAs), and two ribosomal RNA genes (rRNAs)) and a control region [12, 13]. Mitochondrial genomes are widely used in molecular evolution, population genetic structure, biogeography studies, and phylogenetic analysis, due to their small size, stable genetic composition, relatively conservative gene sequence, and complete molecular information [14–17].

Current classification of the tribes and subfamilies of Pentatomidae is based on traditional taxonomic studies. Rider et al. (2018) [2] described each tribe and subfamily of Pentatomidae based on their morphology, providing a good framework for phylogenetic analysis. In recent years, increasing amounts of molecular data on pentatomid species have become available, but most of the studies to date focused on the high-level hierarchical relationships, such as Pentatomoidea or Pentatomomorpha. For example, Yuan et al. (2015) [18] constructed the phylogenetic tree based on a 13 PCGs dataset, which strongly supported the monophyly of Pentatomoidea. Mu et al. (2022) [19] supported this result. Xu et al. (2021) [20] constructed a phylogenetic tree based on PCGRNA and PCG12RNA datasets using 55 species of Pentatomoidea, and resulted that site-heterogeneous mixture models can provide a more stable phylogenetic relationship. Grazia et al. (2008) [21] supported the monophyly of Pentatomidae based on morphological and molecular characteristics, and Zhao (2017) [22] supported this result. In a recent study, Genevcius et al. (2021) [23] used 69 morphological characteristics and five DNA loci to study the phylogeny of Pentatomidae, and reported that most subfamilies and tribes included in Pentatomidae were not monophyletic. Roca-Cusachs et al. (2022) [24] simultaneously rejected the currently accepted monophyletic nature of Pentatomidae. Owing to a lack of robust phylogenetic methods and incomplete sampling, the internal relationships of Pentatomidae remain largely unknown.

We used phylogenetic and molecular clock analyses to explain the origin and evolution of Pentatomidae. Previously, Li et al. (2017) [25] analyzed phylogeny, reconstructed the ancestral characteristic state, and estimated divergence time, indicating that insect diversity may be largely due to coevolution with angiosperms, and key adaptive innovations (such as prognathous mouthpart and predatory behavior) facilitated multiple independent shifts among diverse feeding habits. This study provides a good reference for determining the origin of Pentatomidae. However, no studies have systematically evaluated the divergence time of Pentatomidae; therefore it is particularly important to study the evolution of Pentatomidae by combining fossil data with molecular characteristics.

In this study, we provide a description of the morphological characteristics of *M. indica*, publish a complete mitochondrial genome obtained by high-throughput sequencing, and describe our detailed analyses of mitochondrial genome characteristics. By analyzing codon preference, RNA secondary structure, and evolution rates among Pentatomidae species, we can clarify internal relationships among Pentatomidae. In addition, our results from constructing phylogenetic trees of Pentatomidae and evaluating divergence time will help in understanding Pentatomidae evolution.

## Materials and methods

### Descriptions and measurements

Male genitalia were observed and illustrated after treatment with warm 5% NaOH solution for approximately 20 min. Female genitalia were only illustrated externally. Photographs of both dorsal and ventral habitus were taken using a Nikon SMZ1000 microscope equipped with a computer-controlled SPOT RT digital camera and Helicon software. The terminology used to describe the external genitalia follows that of Fan et al. (2011) [11]. All measurements were performed in millimeters.

Body length was measured from the apex of the head to the tips of the membrane of the hemelytra. Head width was measured between the eyes, and head length was measured from the tip of the head to the midpoint of the anterior margin of the pronotum. Pronotum length was measured from the midpoint of the anterior margin to the midpoint of the posterior margin, and width was measured across the greatest width of the pronotum. Scutellum length was measured from the midpoint of the anterior margin of the scutellum to the apex, and width was measured across the basal angles.

### Sample collection and DNA extraction

Adult *M. indica* specimens were collected from Xiaochantan Wharf (109°10′ E, 19°43′ N), Yangpu Port, Danzhou City, Hainan Province, China, on December 22, 2020. The species we used for scientific purposes is not protected animals and meet animal ethical requirements. It is ethical, humane and responsible. All specimens were immediately placed in absolute ethanol and stored in a freezer at -20°C until DNA extraction. Total DNA was extracted from thoracic tissue using a Genomic DNA Extraction Kit (Sangon Biotech, Shanghai, China).

### Sequencing, assembly, annotation and sequence analyses

A fluorescent dye Quant it PicoGreen dsDNA Assay Kit was used to determine the total amount of DNA. The total amount of DNA was 2.39 μg, and concentration by fluorescence was 47.80 ng/μl. After quality inspection, the required genomic library was constructed using the standard Illumina TruSeq Nano DNA LT library preparation process (Illumina TruSeq DNA Sample Preparation Guide). The mitochondrial genome of *M. indica* was sequenced on an Illumina Novaseq 6000 Platform, using the sequencing mode was paired-end 2 × 150 bp. Fastp v 0.23.1 [26] software was used to filter the original data to obtain high-quality clean data. Geneious v. 11.0 [27] software was used to assemble and annotate the sequences. Reference sequence (*Plautia lushanica* Yang, 1934, NC_058973) [20] for assembly and annotation was obtained from the NCBI databases. The PCGs were edited manually using the open reading frame finder (ORF) (http://www.ncbi.nlm.nih.gov/gorf/gorf.html) with the invertebrate mitochondrial code. The locations of each protein-coding gene's initiation and stop codons were determined by comparison with homologous genes from other insects. MITOS Web (http://mitos.bioinf.uni-leipzig.de/) [28] was used to predict the locations and secondary

structures of the 22 tRNAs. The boundaries of the two rRNAs genes were determined by comparison with those of previously reported mitogenomes. The location of the control region was identified by the boundaries of the neighboring genes.

A circular map of the *M. indica* mitochondrial genome was produced using the CGView Server [29]. Codon usage and nucleotide composition of the PCGs were determined by MEGA v.11.0 [30], and the skew in nucleotide composition was calculated by the following formula: AT-skew = (A − T) / (A + T); GC-skew = (G − C) / (G + C) [31]. Codon W1.4.2 [32] was used to calculate the effective number of codons (ENCs) in the 13 PCGs observed in 50 Pentatomidae species. To study the pattern of evolutionary divergence among the mitochondrial genomes of Pentatomidae species, DnaSP v.6.12.03 [33] was used to count non-synonymous substitutions (Ka) and synonymous substitutions (Ks) in the 13 PCGs of Pentatominae and to calculate Ka/Ks values. In addition, MEGA v.11.0 was used to calculate the conservative sites of tRNA and rRNA genes, and tandem repeats within the control region were identified using the Tandem Repeats Finder server (http://tandem.bu.edu/trf/trf.html) [34].

## Phylogenetic analyses

We selected 50 Pentatomidae species as ingroups (including all available Pentatomidae sequences) and two Scutelleridae species as outgroups to discuss the phylogenetic relationships among the tribes within the family Pentatomidae (Table 1). Phylogenetic relationships were reconstructed based on two datasets: (1) 13 PCGs (2) 13 PCGs + 2 rRNAs + 22 tRNAs (PRT).

The PCGs and RNA genes were extracted using Geneious v.11.0, and MEGA v.11.0 was used to align multiple protein and RNA coding genes sequences. The connection of multiple sequences for each species was achieved using Sequence Matrix v.1.7.8 [35]. Gblocks [36] was used to delete ambiguous sites.

Before constructing a phylogenetic tree, base substitution saturation and sequence composition heterogeneity analyses were performed on both datasets. DAMBE v.7.0.35 software [37] was used to calculate the base substitution saturation index. If Iss < Iss. c indicates that the dataset can be used for phylogenetic analysis. Heterogeneity analysis was performed using Ali-GROOVE v.1.0.8 [38]. Datasets with less heterogeneity were suitable for phylogenetic analysis.

PartitionFinder v.2.1.1 [39] was used to partition models, and alternative models were calculated for each dataset (S1 and S2 Tables). Based on these two datasets, the Bayesian method (BI) was used to reconstruct Pentatomidae phylogenetic trees. BI trees were constructed by MrBayes v.3.2.6 [40]. Two independent runs of 20 million generations were conducted for the matrix, sampling every 1000 generations with a burn-in of 25%. Finally, phylogenetic trees were visualized using the iTOL website (https://itol.embl.de/) [41].

## Divergence time estimate

The relaxation clock lognormal model in BEAST v.1.8.4 [42] was used to estimate Pentatomidae divergence time based on the PCGs dataset. We set up a GTR+I+G partition model using the calibrated Yule model for the prior tree. The fossil information points of Pentatomidae and the genus *Eurydema* Laporte de Castelnau, 1833 [43–45] were used for calibration. Tracer v.1.7.2 [46] was used to confirm the chain convergence. The Markov chain was run twice for every $5{\times}10^8$ generations, sampling every 1000 generations with a burn-in of 25%. The valid sample size for most parameters was greater than 200. Sample trees were aggregated using Tree Annotator v.1.1.8.4, and then 95% highest probability density (95% HPD) was displayed in Figtree v1.4.3 [47].

**Table 1. List of species used to reconstruct the phylogenetic relationships within Pentatomidae.**

| Family | Subfamily | Tribe | Species | GenBank number | Reference |
|---|---|---|---|---|---|
| Pentatomidae | Pentatominae | Antestiini | *Anaxilaus musgravei* | NC_061538 | Unpublished |
| | | Sephelini | *Brachymna tenuis* | NC_042802 | [48] |
| | | Eysarcorini | *Carbula sinica* | NC_037741 | [49] |
| | | Catacanthini | *Catacanthus incarnatus* | NC_042804 | [48] |
| | | Caystrini | *Caystrus obscurus* | NC_042805 | [48] |
| | | Halyini | *Dalpada cinctipes* | NC_058967 | [20] |
| | | Carpocorini | *Dolycoris baccarum* | NC_020373 | [50] |
| | | Halyini | *Erthesina fullo* | NC_042202 | [51] |
| | | Strachiini | *Eurydema dominulus* | NC_044762 | [52] |
| | | Strachiini | *Eurydema gebleri* | NC_027489 | [18] |
| | | Strachiini | *Eurydema liturifera* | NC_044763 | [52] |
| | | Strachiini | *Eurydema maracandica* | NC_037042 | [22] |
| | | Strachiini | *Eurydema oleracea* | NC_044764 | [52] |
| | | Strachiini | *Eurydema qinlingensis* | NC_044765 | [52] |
| | | Strachiini | *Eurydema ventralis* | MG584837 | [52] |
| | | Eysarcorini | *Eysarcoris aeneus* | MK841489 | [53] |
| | | Eysarcorini | *Eysarcoris annamita* | MW852483 | [53] |
| | | Eysarcorini | *Eysarcoris guttigerus* | NC_047222 | [54] |
| | | Eysarcorini | *Eysarcoris montivagus* | MW846867 | [53] |
| | | Eysarcorini | *Eysarcoris rosaceus* | MT165687 | [53] |
| | | Nezarini | *Glaucias dorsalis* | NC_058968 | [20] |
| | | Cappaeini | *Halyomorpha halys* | NC_013272 | [55] |
| | | Caystrini | *Hippotiscus dorsalis* | NC_058969 | [20] |
| | | Hoplistoderini | *Hoplistodera incisa* | NC_042799 | [48] |
| | | Mecideini | *Mecidea indica* | OR654110 | This study |
| | | Menidini | *Menida violacea* | NC_042818 | [48] |
| | | Pentatomini | *Neojurtina typica* | NC_058971 | [20] |
| | | Nezarini | *Nezara viridula* | NC_011755 | [56] |
| | | Nezarini | *Palomena viridissima* | NC_050166 | [57] |
| | | Pentatomini | *Pentatoma metallifera* | NC_058972 | [20] |
| | | Pentatomini | *Pentatoma rufipes* | MT861131 | [58] |
| | | Pentatomini | *Pentatoma semiannulata* | NC_053653 | [59] |
| | | Pentatomini | *Placosternum urus* | NC_042812 | [48] |
| | | Antestiini | *Plautia crossota* | NC_057080 | [60] |
| | | Antestiini | *Plautia fimbriata* | NC_042813 | [48] |
| | | Antestiini | *Plautia lushanica* | NC_058973 | [20] |
| | | Eysarcorini | *Stagonomus gibbosus* | MW846868 | [53] |
| | | Myrocheini | *Tholosanus proximus* | NC_063300 | Unpublished |
| | Phyllocephalinae | Phyllocephalini | *Dalsira scabrata* | NC_037374 | [49] |
| | | Phyllocephalini | *Gonopsis affinis* | NC_036745 | [61] |
| | Podopinae | Deroploini | *Deroploa parva* | NC_063299 | Unpublished |
| | | Graphosomatini | *Graphosoma rubrolineatum* | NC_033875 | Unpublished |
| | | Podopini | *Scotinophara lurida* | NC_042815 | [48] |
| | Asopinae | | *Arma custos* | NC_051562 | [62] |
| | | | *Cazira horvathi* | NC_042817 | [48] |
| | | | *Dinorhynchus dybowskyi* | NC_037724 | [69] |
| | | | *Eocanthecona thomsoni* | NC_042816 | [48] |

*(Continued)*

**Table 1.** (Continued)

| Family | Subfamily | Tribe | Species | GenBank number | Reference |
|---|---|---|---|---|---|
| | | | *Picromerus griseus* | NC_036418 | [63] |
| | | | *Picromerus lewisi* | NC_058610 | [19] |
| | | | *Zicrona caerulea* | NC_058303 | [64] |
| Scutelleridae | Scutellerinae | Scutellerini | *Cantao ocellatus* | NC_042803 | [48] |
| | | Scutellerini | *Chrysocoris stollii* | NC_051942 | Unpublished |

## Results

### Redescription of *Mecidea indica* Dallas, 1851

The body is long and narrow, and dorsum is yellow-white or yellow-brown, mottled with irregular fine dark spots. The venter is yellow-white, with two black longitudinal bands on each lateral side. Light brown punctures are observed on the head and thorax, and punctures are absent or shallow on the abdomen (Fig 1).

The head is triangular, somewhat pointed anteriorly, the juga is longer than the tylus, convergent in front, with straight lateral margin. The eyes are large and prominent, orange, globose, with ocelli located at the posterior margin. Antennae are five-segmented, the first segment is white-yellowish, and does not extend beyond the end of the head; the second segment is extremely long and stout, about twice the length of the third segment, and has three edges, one of which is slightly flattened outward; the remaining segments are cylindrical. The anterior angle of the bucculae protrudes semi-circularly. Its outer margin is relatively straight,

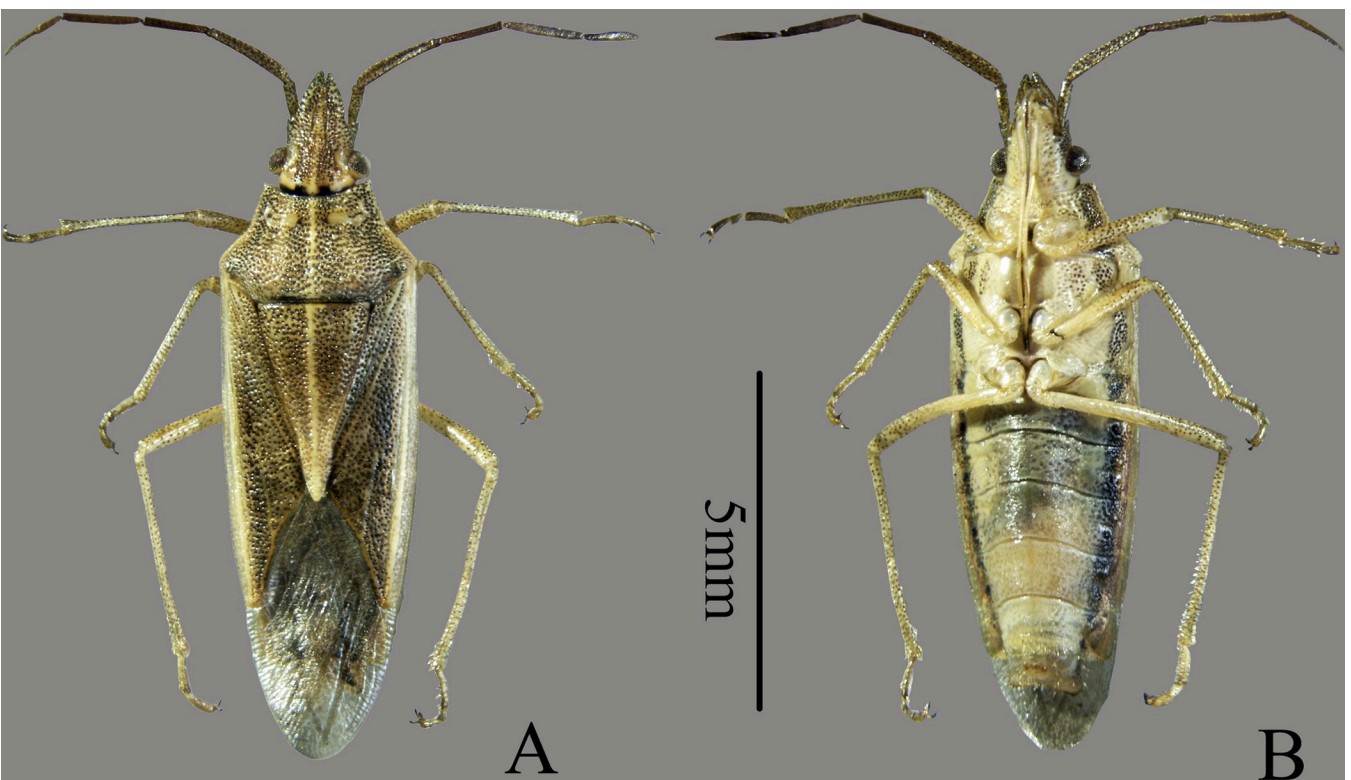

**Fig 1.** Habitus photographs of *Mecidea indica* Dallas, 1851 (A. Dorsal view; B. Ventral view).

and the posterior angle gradually disappears, not exceeding the posterior edge of the eye. The rostrum extends between the mesocoxae and the metacoxae; its first segment does not exceed past the bucculae; the second segment is longer than the two apical segments.

The pronotum is more than three times as long as its wide, its dorsal surface is comparatively flat and coarsely punctured, except for the callus. Humeral angles are round and slightly prominent; anterior angles are short, pointed, and slightly protruding, with their apex flush with the outer margin of the compound eye. The anterior margin is concave, not wider than the distant between eyes, and the posterior margin is straight. The anterior lateral margin is slightly concave, and minutely serrated. The scutellum forms an extremely elongated triangle. Its apical third is yellowish-white, and its apex extends more than half the length of the abdomen. Its lateral margin is narrow with thin edges. The corium is dark, with deep black punctures. The exocorium is usually paler than the corium, yellowish-white, with membrane obviously beyond the abdominal end. A smooth and slightly raised central ridge is longitudinally situated and extends from the base of tylus to the apex of scutellum. The proepisternum is simple; midline of mesosternum is carinate; and the midline of the metasternum is broad with shallowly sulcates. The metathoracic scent gland ostiole extends nearly to the dorsoanterior angle of the pruinose area, its apex sharp. The femora are unarmed, tibiae sulcate, and tarsi 3- segmented, with segment one equivalent to the length of segment two and three. The base half of the claw is yellowish-white, and the apical half is brown.

The abdomen with very shallow or without punctures, two black longitudinal belts are observed on the lateral side. The base of sternite III lacks tubercle. The connexivum are not exposed, and each segment has a black spot around the stoma.

Male genitalia. The pygophore is cup-like, and its width is greater than length, and densely covered with long hair. The posterolateral angles are horned and black; dorsoposterior rim concave and sinuate; and ventroposterior rim have a deep cup-like concave in the middle, with a sharp angle (Fig 2A and 2B). The paramere is simple without any branch, and apex with an elongated black spot (Fig 2C). Aedeagus is simple with one pair of basolateral conjunctival lobes, which apices are not bifurcate but slightly sclerotized, ventral and apical conjunctival lobes haven't been seen; median penial plates strongly sclerotized, united at the base, and distinctly concave apically; vesica protrudes from venter of the median penial plates (Fig 2D).

Female genitalia: The first gonocoxites are large and plate-like, with their inner margins arched and clearly separated. The eighth paratergites are long and oval, with long hair at the apices. The ninth paratergites are also long and oval, with their apices much longer than those of the eighth (Fig 2E).

Material examined. CHINA, Hainan Province: 23 female17 male, Xiaozhaotan Wharf, Yangpu Port, Danzhou City, 22th. XII. 2020, Yuchun Han and RuiMeng leg.

Female measurements: body length 9.8–11.0 mm, width 3.0–3.2 mm; antennal segment length 0.6–0.7: 1.8–2.0: 0.8–1.0:1.1–1.2:1.0; length of head 1.9–2.1 mm, width 1.6–1.8 mm; length of pronotum 1.6–1.8 mm, width 3.0–3.2 mm; length of scutellum 3.3–3.6 mm, width 1.8–2.0 mm.

Male measurements: body length 8.8–9.2 mm, width 2.8–3.0 mm; antennal segment length 0.5–0.7: 1.6–1.7: 0.9–1.1:1.0–1.2:0.9–1.0; length of head 1.8–1.9 mm, width 1.5–1.6 mm; length of pronotum 1.5–1.7 mm, width 2.8–3.0 mm; length of scutellum 3.0–3.2 mm, width 1.6–1.8 mm.

Distribution: China (Hainan, Yunnan, Sichuan); Pakistan; India.

## Mitochondrial genomic structure

The *M. indica* mitochondrial genome is a double stranded circular DNA with a length of 15,670 bp (GenBank accession no. OR654110), containing 37 genes (13 PCGs, 22 tRNA genes,

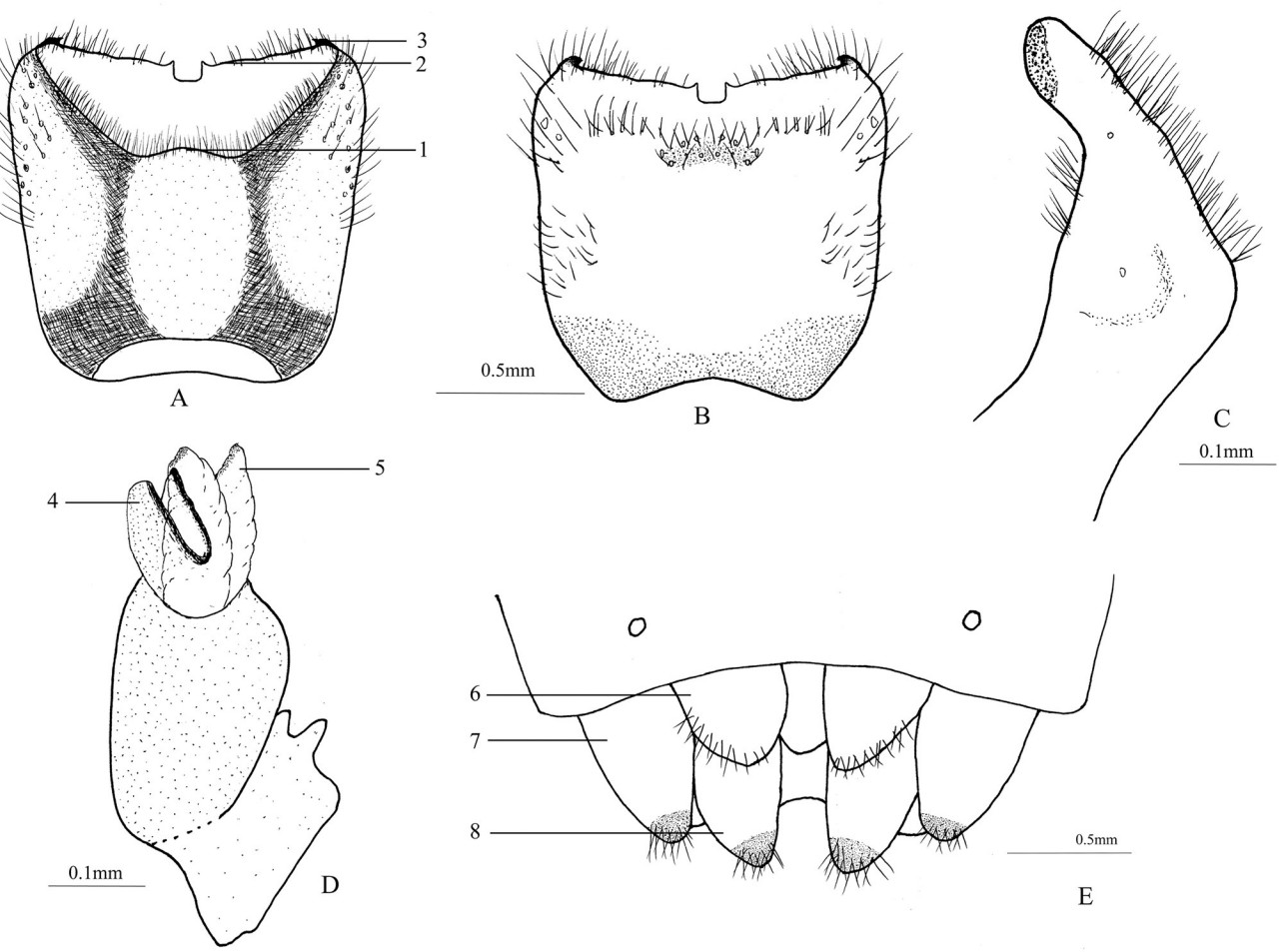

**Fig 2.** A-B. Pygophore (A. Dorsal view; B. Ventral view) (1 dorsoposterior rim; 2 ventroposterior rim; 3 lateroposterior angle). C. Paramere (lateral view). D. Aedeagus (4 median penial plates; 5 basolateral conjunctival lobe). E. Female external genitalia (6 first gonocoxite; 7 eighth paratergite; 8 ninth paratergite).

two rRNA genes), and a control region (Fig 3). The arrangement of the 37 genes is consistent with that of the typical insect *Drosophila yakuba* Burla, 1954, with no gene rearrangement. Fourteen genes are encoded on the N-strand, and 23 genes are encoded on the J-strand (Table 2). The nucleotide composition of the *M. indica* mitochondrial genome is: A (42.97%) >T (33.35%) >C (12.79%) > G (10.89%), AT (76.31%) > GC (23.69%), showing AT-skew and CG-skew (Table 3). The *M. indica* mitochondrial genome contains 15 gene spacers and six gene overlap regions. The gene spacers are 1–24 bp in length, with a total length of 99 bp. The lengths of the overlap regions are 1–8 bp, with a total length of 27 bp. The greatest gene overlap is observed between *trnW* and *trnC*.

## Protein coding genes

The nucleotide composition of the 13 PCGs of *M. indica* is: T (41.76%) >A (33.76%) > G (12.55%) > C (11.93%), AT (75.52%) > GC (24.48%), showing TA-skew and GC-skew. Nine PCGs (*atp6*, *atp8*, *cox1*, *cox2*, *cox3*, *cytb*, *nad2*, *nad3*, and *nad6*) are encoded on the J-strand, and four PCGs (*nad5*, *nad4*, *nad4l*, and *nad1*) are encoded on the N-strand. Most PCGs used ATN (ATT/ATA/ATG) as their initiation codon, except *cox1*, *atp8*, and *nad1*that used TTG as

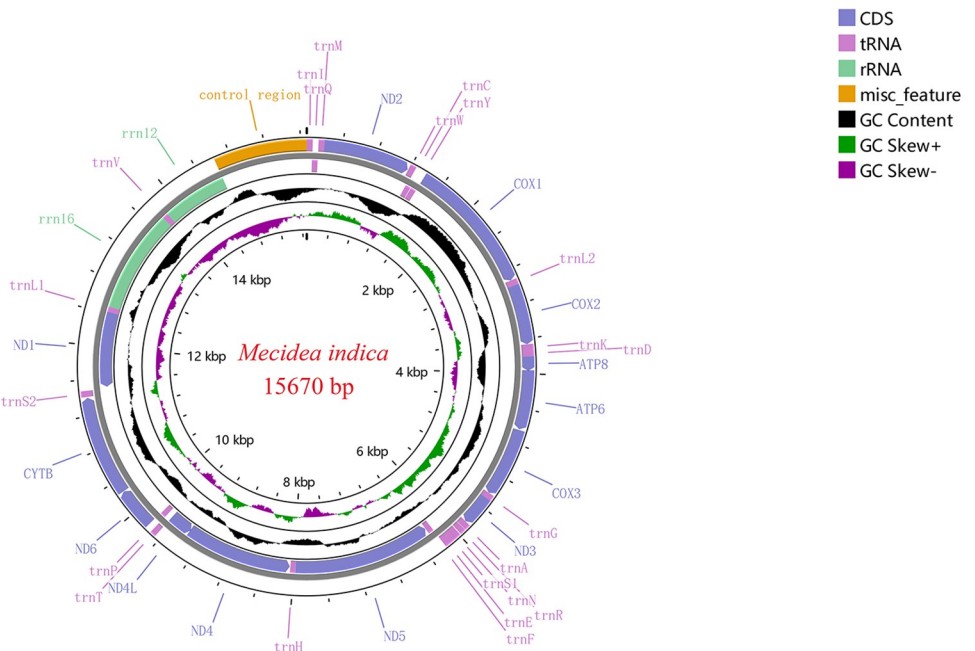

**Fig 3. Mitochondrial genome structure of *M. indica*.**

the initiation codon. The termination codons of most PCGs are TAA, while the termination codon of *cox1* and *cox2* ended with an incomplete T.

Statistics on the relative synonymous codon usage (RSCU) of *M. indica*, the results reveal that the most frequently used codon was UUA (Leu2), while the least commonly used codons were AGG (Ser1) and CCG (Pro) (Fig 4). Among the synonymous codons, those codons ending in A/U at the third base were more frequently used than those codons ending in G/C.

We analyzed the relationship between the effective number of codons (ENC), GC content of all codons, GC content of the first codon position (GC1), GC content of the second codon position (GC2), and GC content of the third codon position (GC3) to further explore the codon usage patterns of Pentatomidae species. The results showed that ENC has a strong positive correlations with GC and GC3 ($R^2 > 0.95$), while ENC has a weak positive correlation with GC1 and GC2 ($R^2 < 0.75$) (Fig 5).

We calculated the synonymous substitution rate (Ks) and non-synonymous substitution rate (Ka) of the PCGs of Pentatomidae. The evolutionary rates of the PCGs are in the order of *atp8 > nad5 > nad4 > nad2 > nad6 > nad4l > nad1 > atp6 > nad3 > cox2 > cox3 > cytb > cox1* (Fig 6). The results showed that Ks>Ka and Ka/Ks<1, indicating that evolution occurred under purifying selection.

## Transfer and ribosomal RNAs

The nucleotide composition of the 22 tRNA genes of *M. indica* was as follows: A (38.62%) > T (37.94%) > G (13.19%) > C (10.25%), AT (76.56%) > GC (23.44%), showing AT-skew and GC-skew. Excepting for *trnS1*, which lacks a stem structure in the DHU arm, all 21 remaining tRNA genes can form a typical cloverleaf structure. The length of the 22 tRNAs was 64–72 bp, with 14 tRNA genes (*trnA, trnD, trnE, trnG, trnI, trnK, trnL2, trnM, trnN, trnR, trnS1, trnS2, trnT,* and *trnW*) on the J-strand, and eight tRNA genes (*trnC, trnF, trnH, trnL1, trnP, trnQ, trnV,* and *trnY*) on the N-strand. Twenty-one wobble G-U pairs, one wobble A-C pair and one

**Table 2. Organization of the mitochondrial genome of *M. indica*.**

| Feature | Strand | Position | Length(bp) | Initiation codon | Stop codon | Anticodon | Intergenic nucleotide |
|---------|--------|----------|-----------|------------------|------------|-----------|----------------------|
| trnI | J | 1–66 | 66 | | | GAT | -3 |
| trnQ | N | 64–132 | 69 | | | TTG | 0 |
| trnM | J | 133–198 | 66 | | | CAT | 0 |
| nad2 | J | 199–1179 | 981 | ATT | TAA | | 7 |
| trnW | J | 1187–1254 | 68 | | | TCA | -8 |
| trnC | N | 1247–1310 | 64 | | | GCA | 6 |
| trnY | N | 1317–1383 | 67 | | | GTA | 1 |
| cox1 | J | 1385–2921 | 1537 | TTG | T | | 0 |
| trnL2 | J | 2922–2988 | 67 | | | TAA | 0 |
| cox2 | J | 2989–3667 | 679 | ATA | T | | 0 |
| trnK | J | 3668–3739 | 72 | | | CTT | 0 |
| trnD | J | 3740–3804 | 65 | | | GTC | 0 |
| atp8 | J | 3805–3966 | 162 | TTG | TAA | | -7 |
| atp6 | J | 3960–4634 | 675 | ATG | TAA | | 9 |
| cox3 | J | 4644–5432 | 789 | ATG | TAA | | -1 |
| trnG | J | 5432–5495 | 64 | | | TCC | 0 |
| nad3 | J | 5496–5846 | 351 | ATA | TAA | | 0 |
| trnA | J | 5847–5913 | 67 | | | TGC | 3 |
| trnR | J | 5917–5980 | 64 | | | TCG | 10 |
| trnN | J | 5991–6058 | 68 | | | GTT | -1 |
| trnS1 | J | 6058–6126 | 69 | | | GCT | 0 |
| trnE | J | 6127–6193 | 67 | | | TTC | 1 |
| trnF | N | 6195–6260 | 66 | | | GAA | 4 |
| nad5 | N | 6265–7974 | 1710 | ATT | TAA | | 1 |
| trnH | N | 7976–8039 | 64 | | | GTG | 8 |
| nad4 | N | 8048–9376 | 1329 | ATG | TAA | | -7 |
| nad4l | N | 9370–9654 | 285 | ATT | TAA | | 2 |
| trnT | J | 9657–9721 | 65 | | | TGT | 0 |
| trnP | N | 9722–9785 | 64 | | | TGG | 10 |
| nad6 | J | 9796–10,260 | 465 | ATA | TAA | | 2 |
| cytb | J | 10,263–11,399 | 1137 | ATG | TAA | | 11 |
| trnS2 | J | 11,411–11,479 | 69 | | | TGA | 24 |
| nad1 | N | 11,504–12,424 | 921 | TTG | TAA | | 0 |
| trnL1 | N | 12,425–12,489 | 65 | | | TAG | 0 |
| rrnL | N | 12,490–13,757 | 1268 | | | | 0 |
| trnV | N | 13,758–13,825 | 68 | | | TAC | 0 |
| rrnS | N | 13,826–14,624 | 799 | | | | 0 |
| CR | J | 14,625–15,670 | 1046 | | | | 0 |

wobble U-C pair were found in 22 tRNAs gene. In Pentatomidae, we observed that 22 tRNA genes contained 41.36% conserved sites (Fig 7).

The nucleotide composition of the *M. indica* rRNA genes is as follows: T (44.46%) > A (34.74%) > G (12.29%) > C (8.51%), and AT (79.20%) > GC (20.80%), showing TA-skew and GC-skew. Both *rrnL* and *rrnS* genes are encoded on the N-strand, with a total length of 2067 bp. In Pentatomidae, *rrnL* contained 35.45% conserved sites and *rrnS* contained 26.37% conserved sites (Figs 8 and 9).

**Table 3. Nucleotide composition and skewness of the mitochondrial genome of *M. indica*.**

| Region | A% | T% | C% | G% | A+T% | G+C% | AT skew | GC skew |
|---|---|---|---|---|---|---|---|---|
| Whole genome | 42.97 | 33.35 | 12.79 | 10.89 | 76.31 | 23.69 | 0.13 | -0.08 |
| PCGs | 33.76 | 41.76 | 11.93 | 12.55 | 75.52 | 24.48 | -0.11 | 0.03 |
| tRNAs | 38.62 | 37.94 | 10.25 | 13.19 | 76.56 | 23.44 | 0.01 | 0.13 |
| rRNAs | 34.74 | 44.46 | 8.51 | 12.29 | 79.20 | 20.80 | -0.12 | 0.18 |
| CR | 37.24 | 40.57 | 13.88 | 8.31 | 77.81 | 22.19 | -0.04 | -0.25 |
| *atp6* | 41.48 | 35.70 | 13.04 | 9.78 | 77.19 | 22.81 | 0.07 | -0.14 |
| *atp8* | 48.77 | 37.65 | 7.41 | 6.17 | 86.42 | 13.58 | 0.13 | -0.09 |
| *cox1* | 34.61 | 34.94 | 14.77 | 15.68 | 69.55 | 30.45 | 0.00 | 0.03 |
| *cox2* | 40.94 | 32.25 | 14.58 | 12.22 | 73.20 | 26.80 | 0.12 | -0.09 |
| *cox3* | 36.38 | 34.98 | 13.81 | 14.83 | 71.36 | 28.64 | 0.02 | 0.04 |
| *nad1* | 26.71 | 50.81 | 9.12 | 13.36 | 77.52 | 22.48 | -0.31 | 0.19 |
| *nad2* | 45.46 | 35.47 | 9.58 | 9.48 | 80.94 | 19.06 | 0.12 | -0.01 |
| *nad3* | 37.89 | 36.47 | 12.25 | 13.39 | 74.36 | 25.64 | 0.02 | 0.04 |
| *nad4* | 24.68 | 51.47 | 11.66 | 12.19 | 76.15 | 23.85 | -0.35 | 0.02 |
| *nad4l* | 26.32 | 48.42 | 9.47 | 15.79 | 74.74 | 25.26 | -0.30 | 0.25 |
| *nad5* | 26.55 | 51.40 | 9.88 | 12.16 | 77.95 | 22.05 | -0.32 | 0.10 |
| *nad6* | 41.51 | 39.57 | 10.11 | 8.82 | 81.08 | 18.92 | 0.02 | -0.07 |
| *cytb* | 34.30 | 38.61 | 14.16 | 12.93 | 72.91 | 27.09 | -0.06 | -0.05 |

## Control region

The control region of *M. indica* is located between *rrnS* and *trnI* (GAT), and is 1046 bp in length. The nucleotide composition of the control region is: T (40.57%)>A (37.24%)>C (13.88%)>G (8.31%), and AT (77.81%)>GC (22.19%), showing TA-skew and CG-skew. We observed eight tandem repeat sequences in the control region with a length range of 18–149 bp (Table 4).

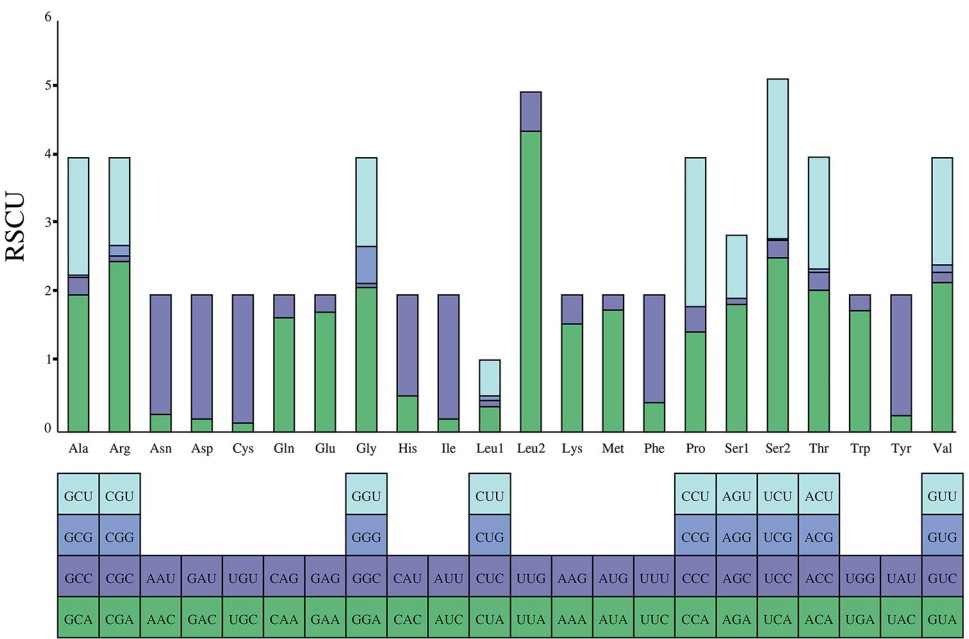

**Fig 4. Relative synonymous codon usage (RSCU) in the mitochondrial genome of *M. indica*.**

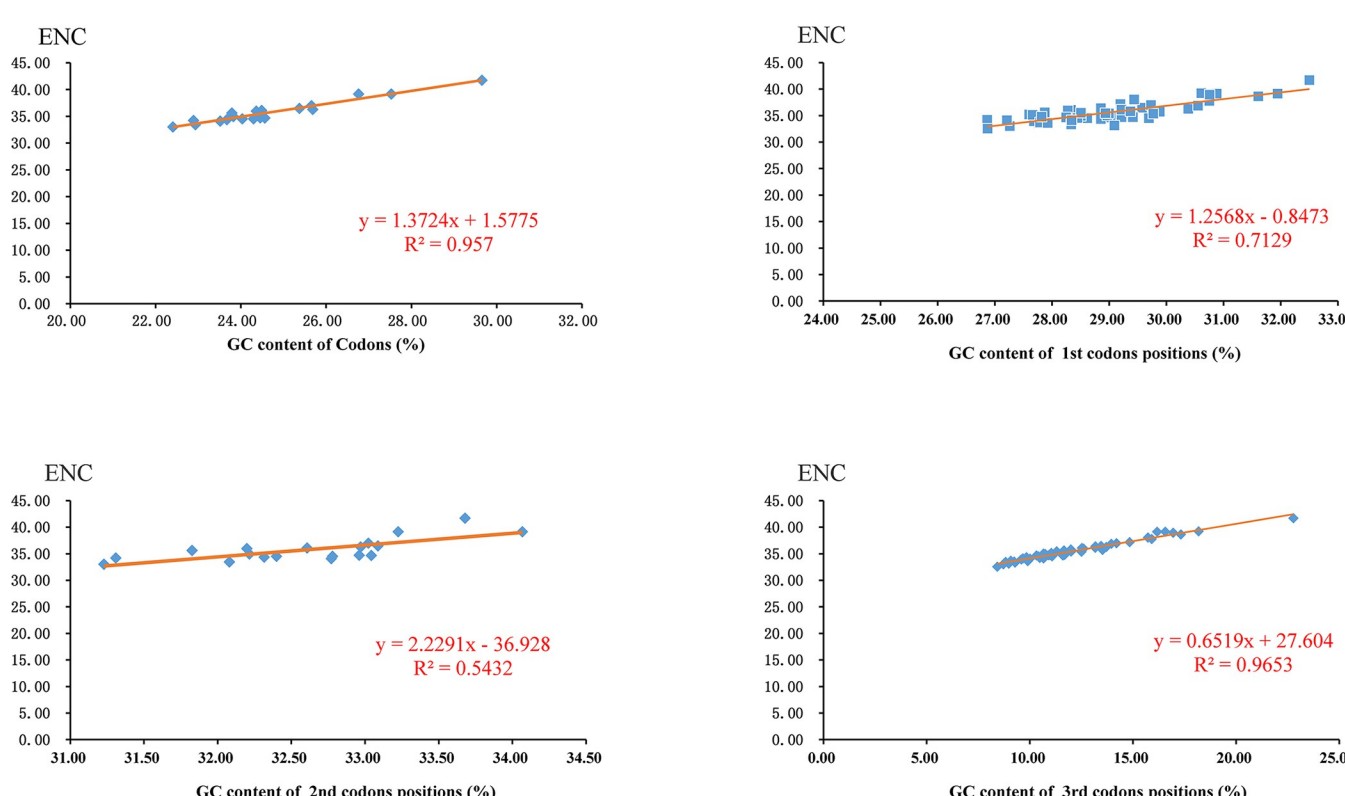

**Fig 5. Evaluation of codon bias in the mitochondrial genomes of Pentatomidae.**

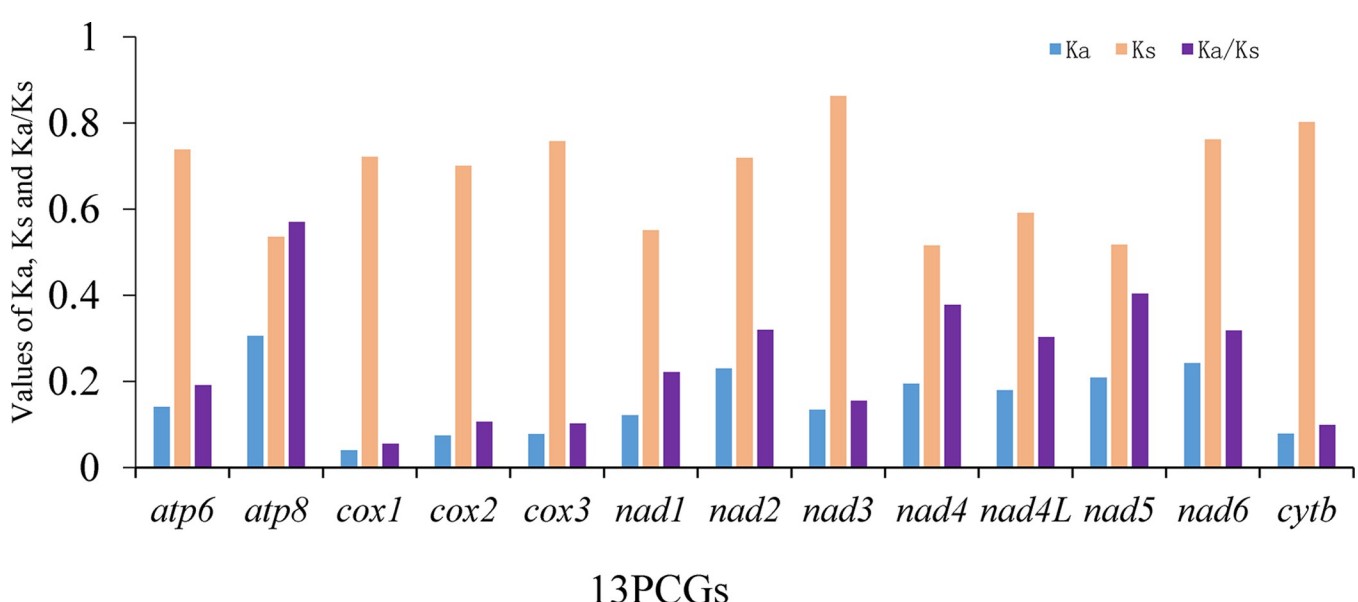

**Fig 6. The Ka, Ks, and Ka/Ks values of 13 PCGs within Pentatomidae.**

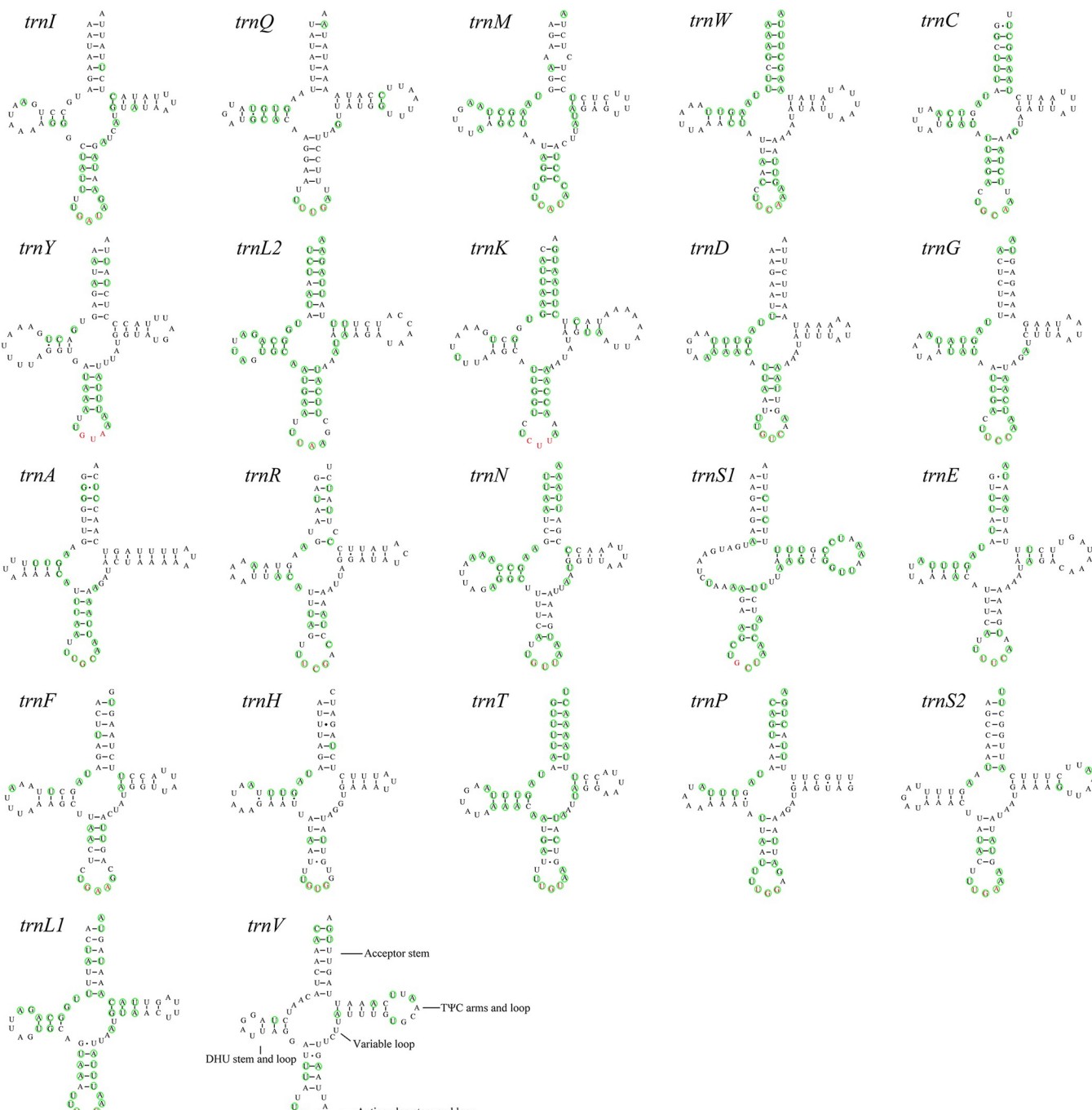

**Fig 7. Potential secondary structure of tRNA in *M. indica*.** Note: The conserved sites within Pentatomidae were marked in green.

## Phylogenetic relationships

Before reconstructing the phylogenetic tree, we performed saturation and heterogeneity analyses on the two datasets (PCGs and PRT). The saturation analysis showed that the sequences of the two datasets are not saturated (Iss<Iss. c, and p<0.05) (Fig 10). Heterogeneity analysis revealed that the composition of the sequences exhibited low heterogeneity (Fig 11). Both studies indicated that these datasets were suitable for phylogenetic studies.

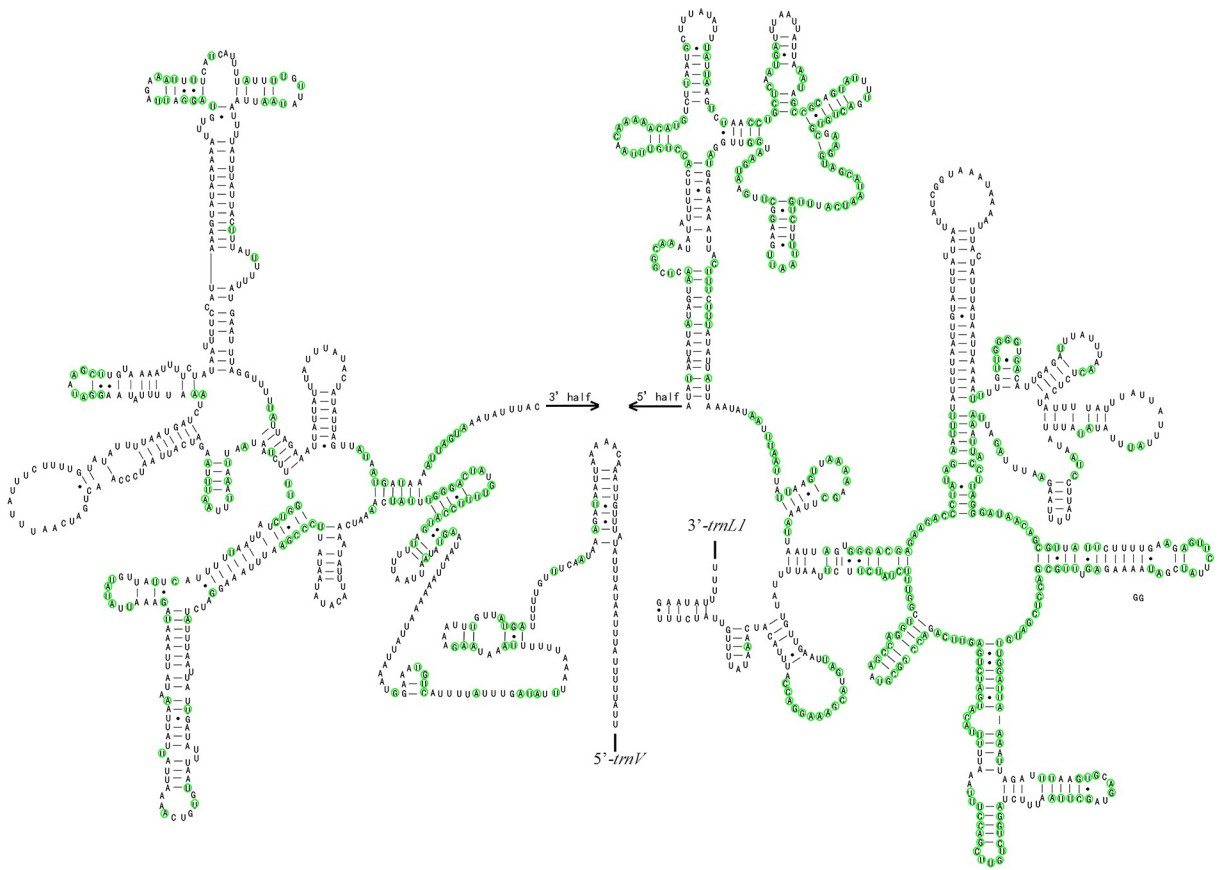

**Fig 8. Potential secondary structure of *rrnL* in *M. indica*.** Note: The conserved sites within Pentatomidae were marked in green.

We constructed phylogenetic trees of Pentatomidae using two datasets (PCGs and PRT) and the Bayesian inference method. The phylogenetic trees based on the each individual datasets had similar topological structures, however, the phylogenetic relationships of *Graphosoma rubrolineatum* (Westwood, 1837) and *Tholosanus proximus* (Dallas, 1851) could not be well determined. Meanwhile, we had selected a phylogenetic tree based on the PRT dataset with the highest bootstrap values to represent our results. The phylogenetic positions of the Pentatominae are as follows: (*Neojurtina* + ((Eysarcorini + (Graphosomatini + (Carpocorini + ((*Palomena* + *Nezara*) + ((*Anaxilaus* + Mecideini) + (*Glaucias* + *Plautia*)))))) + ((((Caystrini + Halyini) + (Cappaeini + (*Placosternum* + Phyllocephalini))) + (Sephelini + (Myrocheini + Deroploini)) + ((Hoplistoderini + (Menidini + Asopinae)) + (*Pentatoma* + ((Podopini + Catacanthini) + Strachiini)))))) (S1 Fig and Fig 12). *Neojurtina typica* Distant, 1921 was the earliest divergent lineage within Pentatomidae. *M. indica* and *Anaxilaus musgravei* Gross, 1976 formed a sister group relationship, Caystrini and Halyini formed a sister group relationship, and strongly supported the monophyly of Strachiini and Eysarcorini.

## Divergence time estimation

We evaluated the divergence time of the Pentatomidae based on the PCGs dataset (Fig 13). The results revealed that the divergence time of the Pentatomidae was 122.75 Mya (95% HPD: 98.76–145.43 Mya), which was in the Aptian stage of the early Cretaceous period

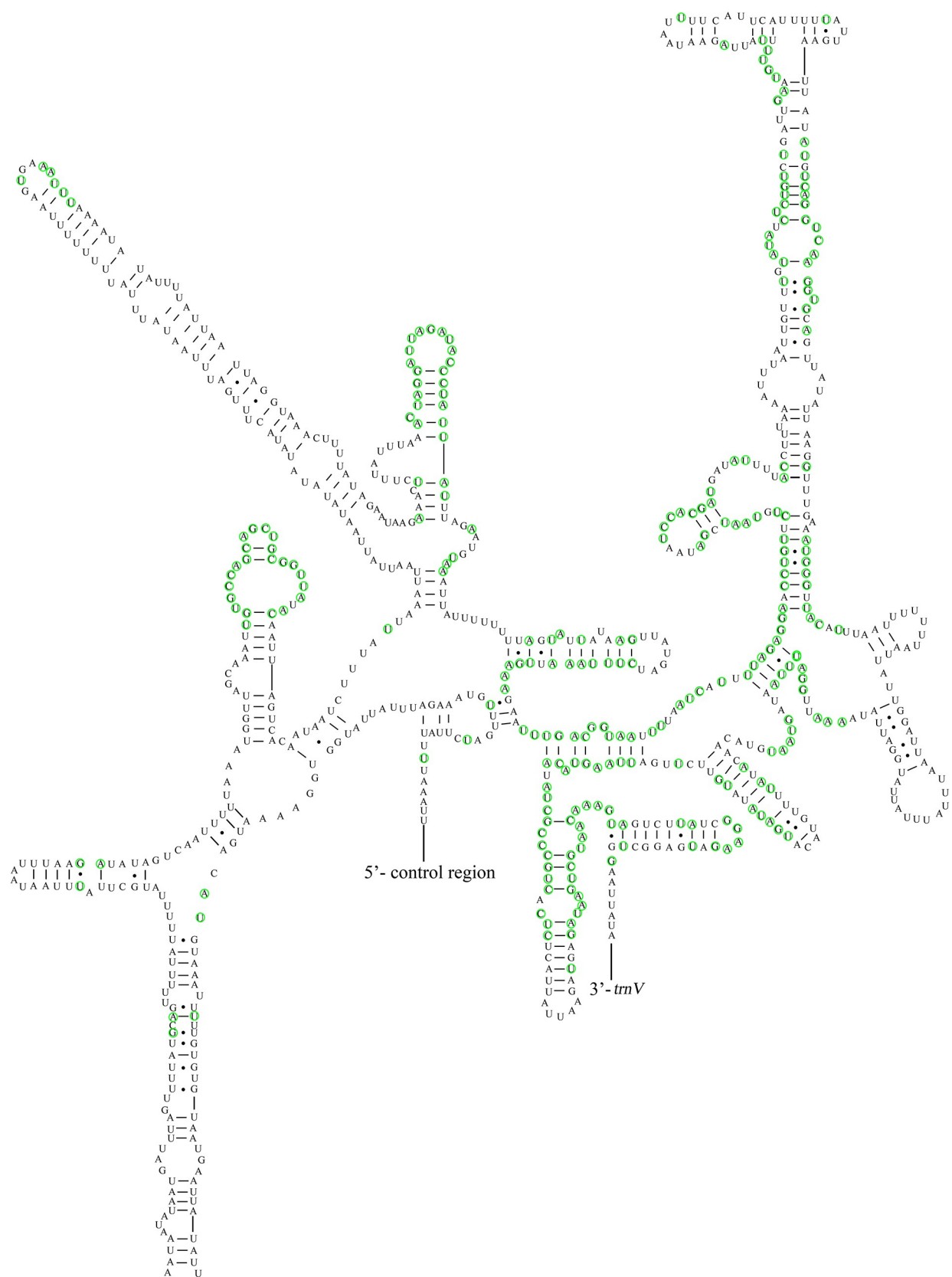

**Fig 9. Potential secondary structure of *rrnS* in *M. indica*.** Note: The conserved sites within Pentatomidae were marked in green.

**Table 4. Tandem repeats of the control region of the mitochondrial genome of *M. indica*.**

| Indices | Period Size | Copy Number | Consensus Size | Percent Matches | Percent Indels | Score | Entropy (0–2) |
|---|---|---|---|---|---|---|---|
| 517–583 | 18 | 3.9 | 18 | 80 | 9 | 79 | 1.57 |
| 723–1046 | 67 | 4.8 | 66 | 86 | 7 | 347 | 1.61 |
| 716–1046 | 33 | 10 | 32 | 75 | 13 | 171 | 1.61 |
| 721–996 | 49 | 5.5 | 49 | 78 | 9 | 231 | 1.62 |
| 723–1028 | 82 | 3.7 | 84 | 79 | 11 | 278 | 1.61 |
| 716–996 | 149 | 1.9 | 149 | 91 | 4 | 451 | 1.61 |
| 750–1046 | 116 | 2.5 | 117 | 83 | 8 | 374 | 1.6 |
| 716–1045 | 149 | 2.2 | 148 | 86 | 7 | 436 | 1.61 |

within the Mesozoic era. Pentatominae and Podopinae are not monophyletic groups, and their phylogenetic relationships are relatively chaotic. As one of the earliest differentiated species in Pentatominae, the divergence time of *N. typica* was 93.59 Mya (95% HPD: 70.94–117.10 Mya) during the Cenomanian stage of the Mesozoic Cretaceous and Late Cretaceous.

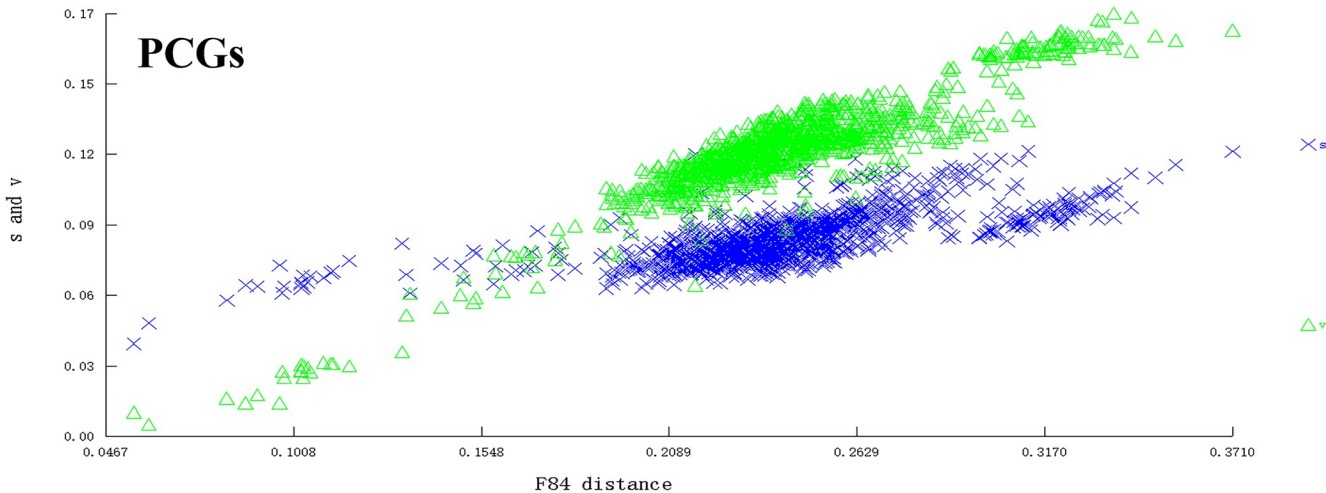

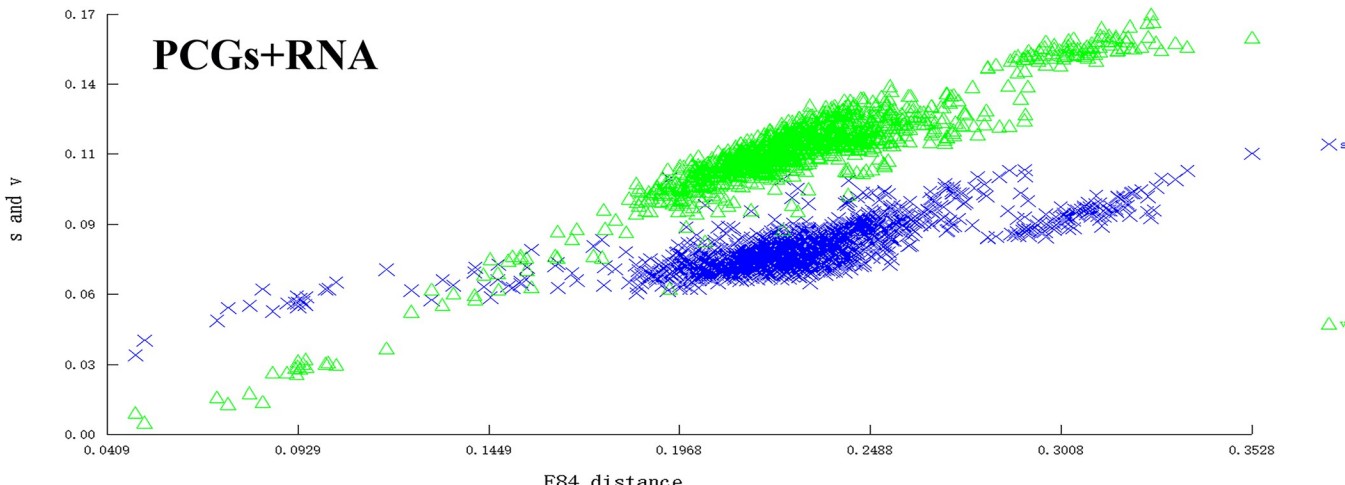

**Fig 10. Saturation analysis based on two datasets (PCGs and PRT).**

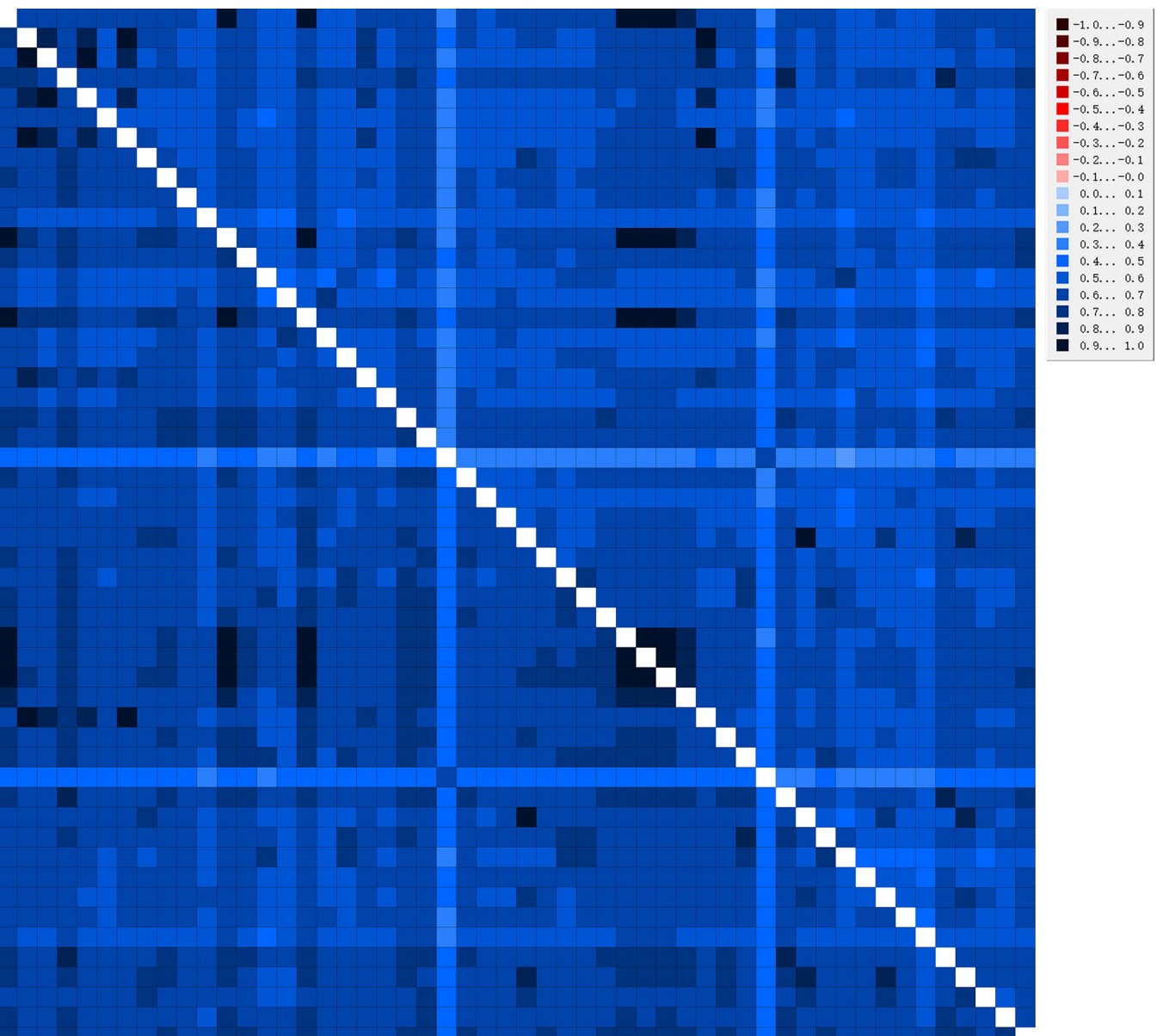

**Fig 11. Heterogeneity analysis based on two datasets (PCGs and PRT).**

The divergence time of *M. indica* and *A. musgravei* was 50.50 Mya (95% HPD: 37.20–64.80 Mya), which was in the Eocene Epulian of Cenozoic Paleogene. In Podopinae, *G. rubrolineatum* was the earliest species to differentiate, with a divergence time of 72.01 Mya (95% HPD: 55.06–90.78 Mya), and was in the Campanian period of the Late Cretaceous of the Mesozoic Cretaceous. The divergence time of Asopinae was 62.32 Mya (95% HPD: 47.08–78.23 Mya), and it was in the Cenozoic Paleogene Paleocene Daning period. The divergence time between two species of the subfamily Phyllocephalinae and *Placosternum urus* Stål, 1876 was 57.75 Mya (95% HPD: 43.12–73.11 Mya), and it was in the Cenozoic Paleogene Paleocene Zanite period.

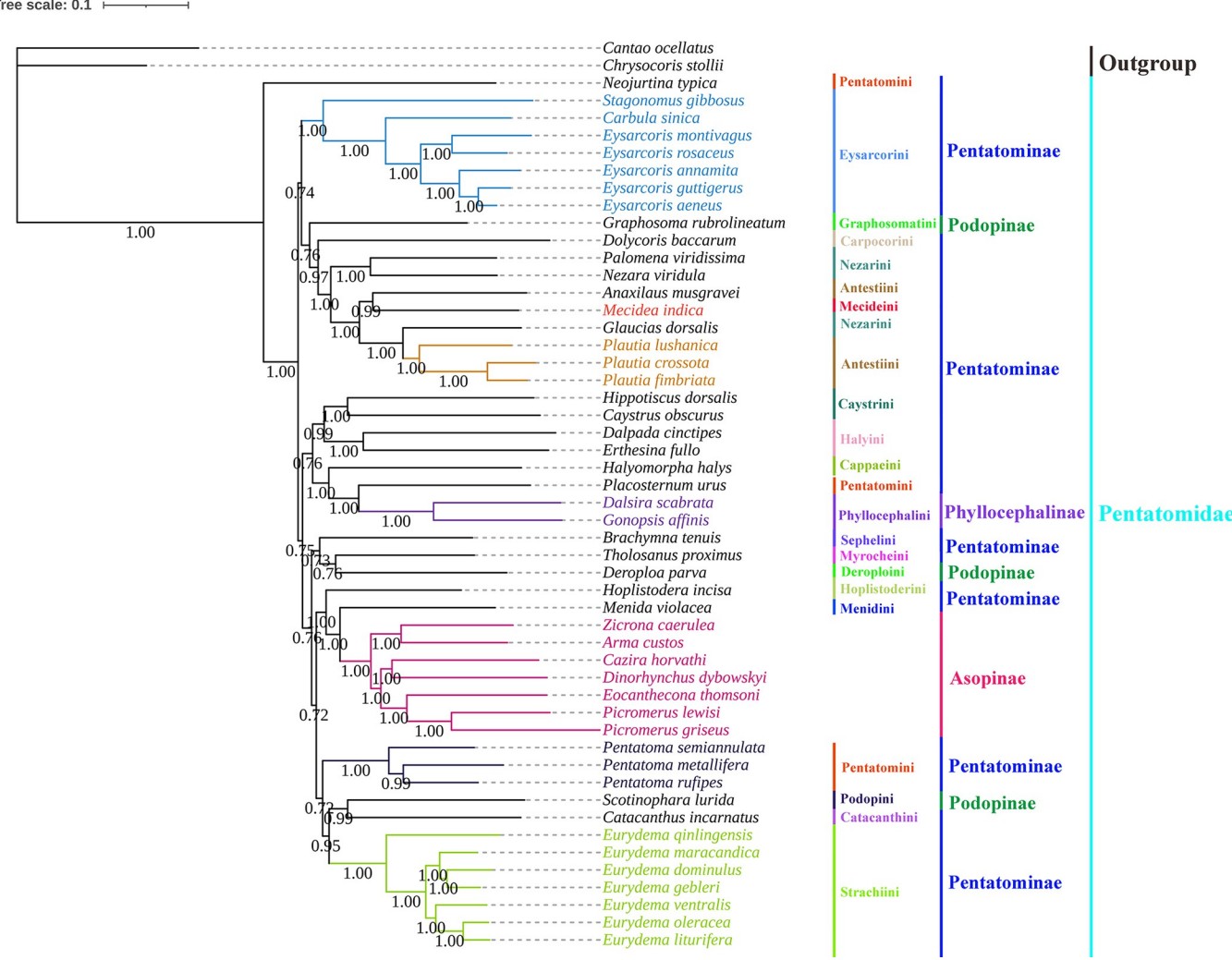

**Fig 12. Phylogenetic tree inferred from PRT constructed using BI analysis.** The number on the branches indicates Bayesian posterior probabilities.

## Discussion and conclusions

In this study, we sequenced the complete mitochondrial genome of *M. indica* using second-generation sequencing technology. The arrangement of the 37 genes was consistent with that of published Pentatomidae species [52, 53, 65], indicating that no gene rearrangements have occurred. The nucleotide composition of the mitochondrial genome of *M. indica* exhibits high AT content, and base composition heterogeneity is common in Heteroptera species [49].

Codon usage bias is a process by which species gradually adapt to their growth environments during evolution. Analyzing codon usage can aid studies of evolution and environmental adaptability of different species. In the *M. indica* mitochondrial genome, we observed a significant AT bias in the nucleotide composition and a preference for codon usage ending with A/T. The evolutionary rate of Pentatomidae was less than one, indicating that they have been subjected to purification selection. The evolution rate of *atp8* was the fastest, whereas that of *cox1* was the slowest, consistent with previous studies [58, 66]. These results indicate that *M. indica* evolution may have been influenced by natural selection.

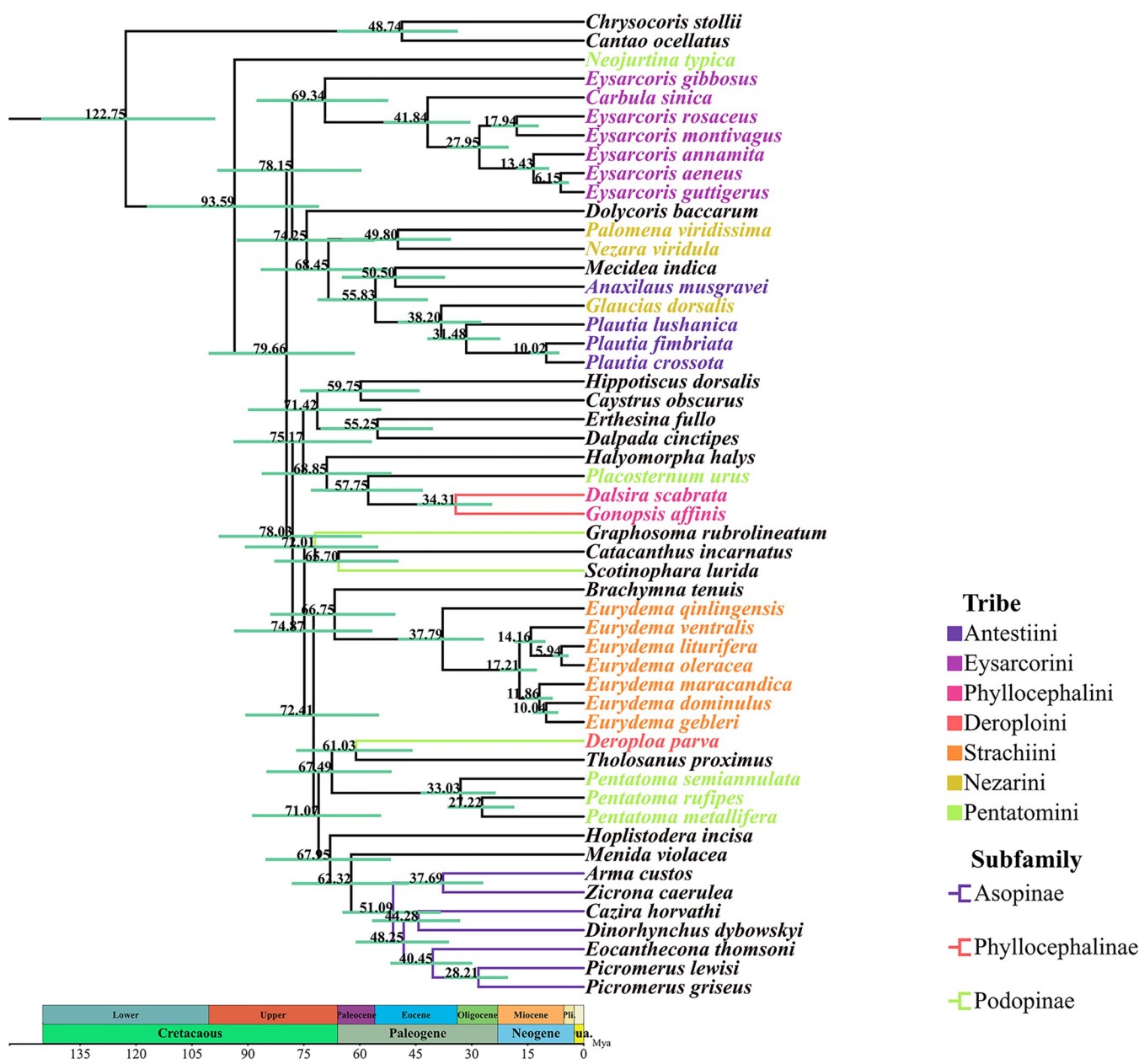

**Fig 13. The chronogram of divergence times by BEAST analysis.**

Except for *trnS1*, the 21 tRNA genes in *M. indica* had typical clover-shaped secondary structures common to many insect species. Some atypical base pairings, such as the G-U pairing, were observed in 22 tRNA genes and two rRNA genes of *M. indica*. These non-Watson-Crick pairings can be converted into fully functional proteins via post-transcriptional mechanisms [67, 68]. The structures of the tRNA genes are more conserved in Pentatomidae than those of the rRNA genes.

The phylogenetic trees we constructed were similar to those constructed via traditional morphological studies (Rider et al. 2018) [2]. *M. indica* has a close genetic relationship with *A. musgravei* and *N. typica* was the first species to differentiate from Pentatomidae. These same

results were obtained by Lian et al. (2022) [66] and Ding et al. (2023) [65]. Our study results rejected the monophyly of Pentatominae and Podopinae, and supported Asopinae as monophyletic. Our results agreed with those of Lian et al. (2022) [66], who supported the monophyletic group of Phyllocephalinae. The monophyly of Eysarcorini and Strachiini is supported in many studies [24, 53, 65]. Halyini and Caystrini are closely related, forming a stable sister group relationship. Ina study by Li et al. (2021) [53], Nezarini and Antestiini were clustered on the same branch, which differs from the results of this study. The relationship between Nezarini and Antestiini remains unclear. In addition, the classification status of Pentatomini, Antestini, and Nezarini was unstable, and more attention should be paid to these tribes in terms of their morphology and molecules. Therefore, more taxa are required to better explain the phylogenetic relationships of the Pentatomidae. The molecular clock method was used to estimate origin and divergence time of each species and to further explore the evolutionary history of Pentatomidae. Pentatomidae species originated in the Cretaceous period of the Mesozoic era, whereas *M. indica* originated in the Paleogene period of the Cenozoic era. In addition, in the evolutionary history of Pentatomidae, a special type of predatory bug has arisen feeding habits have undergone corresponding changes that may be related to environmental changes. This evolutionary history requires further research.

This study is the first to sequence the *M. indica* mitochondrial genome, and provides a theoretical basis for the phylogenetic relationships and evolutionary history of Pentatomidae. Due to the relatively small number of mitochondrial genomes in Pentatomidae, research on the phylogenetic relationships among Pentatomidae is limited and cannot provide good taxonomic position. Therefore, further research is needed to increase the number of mitochondrial genomes in Pentatomidae species and to further elucidate the phylogenetic relationships among Pentatomidae by combining morphological and biological characteristics.

## Supporting information

**S1 Fig. Phylogenetic tree inferred from PCGs constructed using BI analysis.** The number on the branches indicates Bayesian posterior probabilities.
(TIF)

**S1 Table. Partitions and models based on partition finder of PCGs.**
(XLSX)

**S2 Table. Partitions and models based on partition finder of PRT.**
(XLSX)

## Acknowledgments

The authors would like to thank Bo Cai (Hainan Province Engineering Research Center for Quarantine) for providing the specimens, and thanks Editage for linguistic assistance during the preparation and revision of this manuscript.

## Author Contributions

**Conceptualization:** Chao Chen, Xiaofei Ding, Hufang Zhang.

**Data curation:** Shuzhen Yang.

**Formal analysis:** Dongmei Bai.

**Funding acquisition:** Shuzhen Yang, Qing Zhao, Hufang Zhang.

**Investigation:** Dongmei Bai, Zhenhua Zhang.

**Methodology:** Chao Chen.

**Software:** Zhenhua Zhang, Xiaofei Ding, Shuzhen Yang.

**Supervision:** Qing Zhao, Hufang Zhang.

**Visualization:** Zhenhua Zhang.

**Writing – original draft:** Xiaofei Ding.

**Writing – review & editing:** Chao Chen, Qing Zhao.

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
