## [Decision Letter · Decision Letter 0]

20 Dec 2023

PONE-D-23-35518Describe the morphology and mitochondrion genome of  Mecidea indica Dallas, 1851 (Hemiptera, Pentatomidae), with its phylogenetic positionPLOS ONE

Dear Dr. Zhao,

Thank you for submitting your manuscript to PLOS ONE. After careful consideration, we feel that it has merit but does not fully meet PLOS ONE’s publication criteria as it currently stands. Therefore, we invite you to submit a revised version of the manuscript that addresses the points raised during the review process.

We look forward to receiving your revised manuscript.

Kind regards,

Pankaj Bhardwaj, Ph.D.

Academic Editor

PLOS ONE

Comments from Staff Editor: We note that one or more reviewers has recommended that you cite specific previously published works. As always, we recommend that you please review and evaluate the requested works to determine whether they are relevant and should be cited. It is not a requirement to cite these works. We appreciate your attention to this request.

A clean copy of the edited manuscript (uploaded as the new *manuscript* file).

“This research was funded by the National Science Foundation Project of China (No.31872272); the Research Project Supported by Shanxi Scholarship Council of China (Nos. 2020-064), Natural Science Research General Project of Shanxi Province (Nos.202103021224331), Key Forestry Research and Development Plan of Shanxi Province (LYZDYF2023-35).”

Please respond by return e-mail so that we can amend your financial disclosure and competing interests on your behalf.

4. We note that Figures 1 and 2 in your submission contain copyrighted images. All PLOS content is published under the Creative Commons Attribution License (CC BY 4.0), which means that the manuscript, images, and Supporting Information files will be freely available online, and any third party is permitted to access, download, copy, distribute, and use these materials in any way, even commercially, with proper attribution. For more information, see our copyright guidelines: http://journals.plos.org/plosone/s/licenses-and-copyright.

1. You may seek permission from the original copyright holder of Figures 1 and 2 to publish the content specifically under the CC BY 4.0 license.

Reviewers' comments:

Reviewer's Responses to Questions

**Comments to the Author**

1. Is the manuscript technically sound, and do the data support the conclusions?

Reviewer #1: Yes

Reviewer #2: Yes

Reviewer #3: Yes

2. Has the statistical analysis been performed appropriately and rigorously? 

Reviewer #1: Yes

Reviewer #2: Yes

Reviewer #3: Yes

3. Have the authors made all data underlying the findings in their manuscript fully available?

Reviewer #1: Yes

Reviewer #2: Yes

Reviewer #3: Yes

4. Is the manuscript presented in an intelligible fashion and written in standard English?

Reviewer #1: Yes

Reviewer #2: Yes

Reviewer #3: Yes

5. Review Comments to the Author

Reviewer #1: The manuscript have describe the morphology and sequencing the mitochondrion genome of Mecidea indica Dallas,1851 and explored its phylogenetic position. The manuscript is fitted to Plos one and some minor suggestion as follow:

1.The discussion may be strength

2.The language may be improved by a native English speaker

3.Evidence of animal ethics needs to be added in manuscript

4.The figures is not clear

5.Line 80 some new paper you can cite such as :https://doi.org/10.1038/s41598-022-17814-8; DOI: 10.3724/ahr.2095-0357.2023.0003

Reviewer #2: The aim of the study is to sequence a mitochondrion genome of Mecidea indica, to perfom a phylogenetic analysis of the family Pentatomidae, as well as an divergence time estimation on this phylogeny using fossil data. In recent years, there have been many papers on mitochondrial genome, but few studies have combined morphological description with mitochondrial genome and explored its taxonomic status, so it provides a new integrated classification method.

But there are some problems that the author needs to check:

1. In this paper, the authors checked a series of speicmens, so the species measurements should be a range rather than a fixed value.

2. In “Materials and Methods”, What was the concentration of DNA? Did you use total DNA for sequencing? How the libraries were prepared and how the quality of the libraries was estimated? How did you perform de novo genome assembly in Genious and how did you determined the mitochondrial contigs?

3. It needs a scale bar on the Figure 1 about the two adult figures.

4. Instead of providing many trees, please give one which summarize all your conclusions. All other trees as well as many other graphical data from your research could be given in Supplement.

5. The description of genes should be consistent throughout the text, such as cob in some places and cytb in others.

6. Which model was determined by PartitionFinder? Is this model implemented in MrBayes?

7. The writing and language is not up to standard. Please proofread and use the spell checking functions in the word processing software of choice before submitting manuscripts.

Reviewer #3: In this study, the author described external morphology and the complete mitochondrial genome characteristics of Mecidea indica. Moreover, the author clarified the evolutionary rate and divergence time of M. indica and discussed the phylogenetic relationships of tribes within the family Pentatomidae. At present, the classification of tribes and subfamilies of Pentatomidae is still based on traditional taxonomic studies, this study is important for reconstructing the phylogeny of Pentatomidae and understanding the evolutionary history of Pentatomidae.

6. PLOS authors have the option to publish the peer review history of their article (what does this mean?). If published, this will include your full peer review and any attached files.

Reviewer #1: No

Reviewer #2: No

Reviewer #3: **Yes: **Baoying Guo

---

## [Author Response · Author response to Decision Letter 0]

15 Jan 2024

Dear Editor, 

Thanks for your letter and for reviewers’ comments concerning our manuscript entitled “Describe the morphology and mitochondrion genome of Mecidea indica Dallas, 1851 (Hemiptera, Pentatomidae), with its phylogenetic position”. These comments are all valuable and helpful for revising and improving our paper. We have studied all comments carefully, and have made conscientious correction. Revised portion are marked in red in the paper. The main corrections in the paper and the responds to the reviewers’ comments are as flowing.

Thank you for your consideration. I look forward to hearing from you.

Sincerely

Qing Zhao

Response to Reviewer #1

Point 1: The discussion may be strength.

Response: Thank you very much for your comments on my article, we have made corresponding modifications. We have revised the Discussion accordingly.

Point 2: The language may be improved by a native English speaker.

Response: We have improved our manuscript by Editage service, and the polishing proof of this manuscript is provided.

Point 3: Evidence of animal ethics needs to be added in manuscript.

Response: The species we used for scientific purposes is not protected animals and meet animal ethical requirements. It is ethical, humane and responsible. And we have added the sentence in our manuscript.

Point 4: The figures are not clear.

Response: Thank you very much for your comments on my article. We have processed the figures.

Point 5: Line 80 some new paper you can cite such as: https://doi.org/10.1038/s41598-022-17814-8;

DOI: 10.3724/ahr.2095-0357.2023.0003.

Response: Thank you very much for your comments on my article, we have made corresponding modifications.

Response to Referee #2

Point 1: In this paper, the authors checked a series of specimens, so the species measurements should be a range rather than a fixed value.

Response: You are right that the measurement should be a numerical interval, which we have already added.

Point 2: In “Materials and Methods”, What was the concentration of DNA? Did you use total DNA for sequencing? How the libraries were prepared and how the quality of the libraries was estimated? How did you perform de novo genome assembly in Genious and how did you determined the mitochondrial contigs?

Response: Thank you very much. We used fluorescent dye (Quant it PicoGreen dsDNA Assay Kit) to detect the total amount of DNA. The total amount of DNA was 2.39 g, and the fluorescence concentration was 47.80 ng/l. After passing the quality inspection, the required genomic library was constructed using the standard Illumina TruSeq Nano DNA LT library preparation experimental process (Illumina TruSeq DNA Sample Preparation Guide). We chose “Map to Reference” for assembly instead of “De Novo Assemble”. And we used “HQ%” to determine the mitochondrial contigs. 

Point 3: It needs a scale bar on the Figure 1 about the two adult figures.

Response: Thank you for your advice, we have added the scale bar. 

Point 4: Instead of providing many trees, please give one which summarize all your conclusions. All other trees as well as many other graphical data from your research could be given in Supplement.

Response: Thank you very much for your comments on my article. We had chosen a phylogenetic tree based on the PRT dataset with higher bootstrap values to represent our results. And the other phylogenetic tree was placed in Supplement.

Point 5: The description of genes should be consistent throughout the text, such as cob in some places and cytb in others.

Response: Thank you very much for your comments on my article, we have made corresponding modifications.

Point 6: Which model was determined by PartitionFinder? Is this model implemented in MrBayes?

Response: Thank you very much for your comments on my article. The model of Partition Finder was placed in Supplement. The model is implemented in MrBayes.

Point 7: The writing and language is not up to standard. Please proofread and use the spell checking functions in the word processing software of choice before submitting manuscripts.

Response: Thank you, we have accepted the advice and corrected it. 

Response to Referee #3

Reviewer #3: In this study, the author described external morphology and the complete mitochondrial genome characteristics of Mecidea indica. Moreover, the author clarified the evolutionary rate and divergence time of M. indica and discussed the phylogenetic relationships of tribes within the family Pentatomidae. At present, the classification of tribes and subfamilies of Pentatomidae is still based on traditional taxonomic studies, this study is important for reconstructing the phylogeny of Pentatomidae and understanding the evolutionary history of Pentatomidae.

Response: Thank you very much for your comments on our manuscript.

---

## [Decision Letter · Decision Letter 1]

31 Jan 2024

PONE-D-23-35518R1Describe the morphology and mitochondrion genome of  Mecidea indica Dallas, 1851 (Hemiptera, Pentatomidae), with its phylogenetic positionPLOS ONE

Dear Dr. Zhao,

Thank you for submitting your manuscript to PLOS ONE. After careful consideration, we feel that it has merit but does not fully meet PLOS ONE’s publication criteria as it currently stands. Therefore, we invite you to submit a revised version of the manuscript that addresses the points raised during the review process.

There are a few minor changes that authors need to address. Please submit the revised manuscript so I will make the appropriate decision. 

We look forward to receiving your revised manuscript.

Kind regards,

Pankaj Bhardwaj, Ph.D.

Academic Editor

PLOS ONE

Journal Requirements:

Reviewers' comments:

Reviewer's Responses to Questions

**Comments to the Author**

1. If the authors have adequately addressed your comments raised in a previous round of review and you feel that this manuscript is now acceptable for publication, you may indicate that here to bypass the “Comments to the Author” section, enter your conflict of interest statement in the “Confidential to Editor” section, and submit your "Accept" recommendation.

Reviewer #2: All comments have been addressed

Reviewer #3: All comments have been addressed

2. Is the manuscript technically sound, and do the data support the conclusions?

Reviewer #2: Yes

Reviewer #3: Yes

3. Has the statistical analysis been performed appropriately and rigorously? 

Reviewer #2: Yes

Reviewer #3: Yes

4. Have the authors made all data underlying the findings in their manuscript fully available?

Reviewer #2: Yes

Reviewer #3: Yes

5. Is the manuscript presented in an intelligible fashion and written in standard English?

Reviewer #2: Yes

Reviewer #3: Yes

6. Review Comments to the Author

Reviewer #2: (No Response)

Reviewer #3: Point 1: The format of the numbers should be consistent, for example: “15,670 bp” on line 33 is different from “15670 bp”on line 323.

Point 2: In line 202, “95%HPD” should be “95% HPD”.

Point 3: The abbreviation "protein-coding genes (PCGs)" should appear when "protein-coding genes" first appears, and it should not be repeated later, for example: in line 174 and line 324.

7. PLOS authors have the option to publish the peer review history of their article (what does this mean?). If published, this will include your full peer review and any attached files.

Reviewer #2: No

Reviewer #3: **Yes: **Baoying Guo

---

## [Author Response · Author response to Decision Letter 1]

7 Feb 2024

Dear Editor, 

Thanks for your letter and for reviewers’ comments concerning our manuscript entitled “Describe the morphology and mitochondrion genome of Mecidea indica Dallas, 1851 (Hemiptera, Pentatomidae), with its phylogenetic position”. These comments are all valuable and helpful for revising and improving our paper. We have studied all comments carefully, and have made conscientious correction. Revised portion are marked in red in the paper. The main corrections in the paper and the responds to the reviewers’ comments are as flowing.

Thank you for your consideration. I look forward to hearing from you.

Sincerely

Qing Zhao

Response to Reviewer #3

Point 1: The format of the numbers should be consistent, for example: “15,670 bp” on line 33 is different from “15670 bp”on line 323.

Response: Thank you very much for your comments on my article, we have made corresponding modifications. 

Point 2: Point 2: In line 202, “95%HPD” should be “95% HPD”.

Response: We have improved our manuscript by Editage service, and the polishing proof of this manuscript is provided.

Point 3: The abbreviation "protein-coding genes (PCGs)" should appear when "protein-coding genes" first appears, and it should not be repeated later, for example: in line 174 and line 324.

Response: Thank you very much for your comments on my article, we have made corresponding modifications.

---

## [Editor Report · Decision Letter 2]

8 Feb 2024

Describe the morphology and mitochondrion genome of  Mecidea indica Dallas, 1851 (Hemiptera, Pentatomidae), with its phylogenetic position

PONE-D-23-35518R2

Dear Dr. Zhao,

We’re pleased to inform you that your manuscript has been judged scientifically suitable for publication and will be formally accepted for publication once it meets all outstanding technical requirements.

Kind regards,

Pankaj Bhardwaj, Ph.D.

Academic Editor

PLOS ONE
---

## [Editor Report · Acceptance letter]

18 Mar 2024

PONE-D-23-35518R2 

PLOS ONE

Dear Dr. Zhao, 

I'm pleased to inform you that your manuscript has been deemed suitable for publication in PLOS ONE. Congratulations! Your manuscript is now being handed over to our production team.

Kind regards, 

on behalf of

Dr. Pankaj Bhardwaj 

Academic Editor

PLOS ONE